# Suppression of RBFox2 by Multiple MiRNAs in Pressure Overload-Induced Heart Failure

**DOI:** 10.3390/ijms24021283

**Published:** 2023-01-09

**Authors:** Mingyao Gu, Yuying Zhao, Hong Wang, Wanwen Cheng, Jie Liu, Kunfu Ouyang, Chaoliang Wei

**Affiliations:** 1Shenzhen Key Laboratory of Metabolism and Cardiovascular Homeostasis, Health Science Center, Shenzhen University, Shenzhen 518060, China; 2Department of Cardiovascular Surgery, Peking University Shenzhen Hospital, Shenzhen 518036, China; 3Department of Pathophysiology, Health Science Center, Shenzhen University, Shenzhen 518060, China

**Keywords:** RBFox2, miRNAs, E–C coupling, alternative splicing, heart failure

## Abstract

Heart failure is the final stage of various cardiovascular diseases and seriously threatens human health. Increasing mediators have been found to be involved in the pathogenesis of heart failure, including the RNA binding protein RBFox2. It participates in multiple aspects of the regulation of cardiac function and plays a critical role in the process of heart failure. However, how RBFox2 itself is regulated remains unclear. Here, we dissected transcriptomic signatures, including mRNAs and miRNAs, in a mouse model of heart failure after TAC surgery. A global analysis showed that an asymmetric alternation in gene expression and a large-scale upregulation of miRNAs occurred in heart failure. An association analysis revealed that the latter not only contributed to the degradation of numerous mRNA transcripts, but also suppressed the translation of key proteins such as RBFox2. With the aid of Ago2 CLIP-seq data, luciferase assays verified that RBFox2 was targeted by multiple miRNAs, including Let-7, miR-16, and miR-200b, which were significantly upregulated in heart failure. The overexpression of these miRNAs suppressed the RBFox2 protein and its downstream effects in cardiomyocytes, which was evidenced by the suppressed alternative splicing of the Enah gene and impaired E–C coupling via the repression of the Jph2 protein. The inhibition of Let-7, the most abundant miRNA family targeting RBFox2, could restore the RBFox2 protein as well as its downstream effects in dysfunctional cardiomyocytes induced by ISO treatment. In all, these findings revealed the molecular mechanism leading to RBFox2 depression in heart failure, and provided an approach to rescue RBFox2 through miRNA inhibition for the treatment of heart failure.

## 1. Introduction

Heart failure, the final common fate of numerous cardiac diseases [1], has always been a focal issue in biomedical research. Generally, the development of cardiac disease has a process, from the compensatory response in the early stage to the compensatory imbalance in the later stages, and it eventually develops into heart failure [2]. Cardiac hypertrophy is not a prerequisite for heart failure, although it has been widely recognized as a compensatory program in pathological conditions [3]. Myocardial insufficiency, resulting from the decompensation process induced by sustained chronic stress or disease, is key in the progression to heart failure [4]. To discover biomarkers, genomics approaches have been applied to heart failure samples from both humans and animals over recent years [5,6,7]. Numerous genes affecting transcription, the cytoskeleton, calcium handling, signal transudation, and inflammation have been identified to be involved in the pathogenesis of heart failure. However, further evidence is required to prove whether these genes can be applied as biomarkers for diagnoses or therapy to prevent the progression of heart failure.

MicroRNAs (miRNAs) are small, non-coding RNAs that are incorporated with the RNA-induced silencing complex (RISC) to stimulate mRNA degradation or inhibit protein translation [8]. Argonaute 2 (Ago2) is a core element of the RISC that mediates miRNA-guided gene-silencing processes. Increasing evidence suggests that a large number of miRNAs are involved in heart failure at the post-transcriptional level [9]. A small number of miRNAs have been characterized in detail with reported targets in cardiac disease, suggesting their potential to serve as biomarkers for heart failure. MiR-1, the most abundant miRNA in heart tissue, targets the key cardiac-specific transcription factors Hand2 and Irx-5, which are critical for cardiogenesis and cardiac conduction [10]. Furthermore, miR-1 targets the 3′UTRs of CaM, Mef2a, and Gata4 to regulate cardiomyocyte growth responses [11]. MiR-208a and miR-208b have been found to directly target Thrap1 and myostatin, which act as negative regulators for cardiac growth and hypertrophy [12]. MiR-22 directly inhibits a large number of transcription factors, such as Purb, Hdac4, PGC-1α, PPARα, and SIRT1, resulting in cardiac contractile and calcium-handling dysfunction in mice [13]. Although these miRNAs were verified to repress multiple genes related to cardiac function, many other critical genes have yet to be associated with cardiac-specific miRNAs that target to them. More mechanistic studies from the perspective of the genes themselves are required to reveal how those critical cardiac genes are regulated by miRNAs.

Our recent research on the RNA binding protein RBFox2 (also known as RMB9, Fox2, or Fxh) has proven its functional importance in maintaining cardiac performance and playing a key role in the process of heart failure [14,15,16]. RBFox2 is expressed from embryo to adult and is greatly reduced in mouse heart failure models [14,17]. RbFox2 heart-specific knockout mice showed an obvious phenotype of heart failure two months after birth, including decreased heart ejection, reduced cardiac contractility, and impaired calcium handling [14,16]. However, a cardiac-specific KO of RBFox2 did not produce a cardiac hypertrophy phenotype during development, indicating that its knockout directly induced a decompensation process leading to heart failure. Mechanistically, RBFox2 regulates numerous altered splicing events linked to specific heart failure phenotypes, which may constitute a key part of the developmental program during postnatal heart remodeling. A new function of RBFox2 in regulating transcriptional repression was found in several cell types, including cardiomyocytes, which can recruit polycomb complex 2 (PRC2) in a nascent RNA-dependent manner [15]. A further mechanistic study revealed that a cardiac ablation of the RBFox2 protein resulted in transcriptionally induced miRNAs in combination with enhanced miRNAs targeting Jph2, which directly contributed to heart failure [16]. In addition to our work, the results from other groups have confirmed that RBFox2 contributes to heart development and is required for maintaining normal function of the human heart, as mutations in or deletions of the RBFox2 protein cause defective formation of the cardiac chambers and various congenital or acquired heart diseases [18,19,20,21]. All these relative results suggest that the RBFox2 protein is a key cardioprotective factor with therapeutic potential for treating heart failure. However, it remains unclear how RBFox2 itself is regulated in heart failure.

In the present study, we dissected the transcriptomic signature, including mRNAs and miRNAs, of isolated cardiomyocytes from a mouse model of heart failure. Global and association analyses suggested that the large-scale upregulation of miRNAs contributed to the asymmetric changes in gene expression in heart failure. Importantly, RBFox2 proved to be targeted by multiple upregulated miRNAs that impaired both the RBFox2 protein and its downstream functions. The inhibition of corresponding miRNAs could rescue the RBFox2 protein as well as its downstream effects in dysfunctional cardiomyocytes. These findings revealed the molecular mechanism for how cardioprotective factor RBFox2 is repressed in heart failure and provided a possible approach for restoring cardiac function by rescuing RBFox2 through miRNA inhibition.

## 2. Results

### 2.1. Identification of Transcriptomic Signatures of Cardiomyocytes in TAC Heart Failure Model

Transverse aortic constriction (TAC) is a well-established mouse model for inducing heart failure caused by a pressure overload [22]. Echocardiographic results showed an obvious heart failure phenotype in TAC mouse hearts at 14 weeks, which was characterized by a significant increase in both the end-diastolic and end-systolic left ventricular internal dimensions (LVIDs and LVIDd) (Figure 1A,B), and a great decrease (~50%) in the fractional shortening (FS) and ejection fraction (EF) (Figure 1C,D). In addition, the ventricle weight (VW) and lung weight (LW) increased while body weight remained unchanged (Appendix A) after TAC surgery, which resulted in a significant increase in both the ventricle weight and lung weight ratio to body weight (VW/BW and LW/BW) (Figure 1E,F). These results indicate that long-term TAC surgery leads to cardiac remodeling and progression to heart failure.

The profiling of mRNA was performed in duplicate to detect global gene expression changes in isolated cardiomyocytes from the 14-week hearts of TAC or sham mice (Appendix A). Among 8163 of relatively highly expressed genes (FPKM > 2) (Appendix A), 149 genes showed significant changes in the TAC group by a strict cut-off (*p* < 0.05; fold change > 2.0) (Figure 1G and Appendix A). These dysregulated genes included heart failure biomarkers such as *Nppa*, *Nppb*, and *Gdf15*, which are wildly used in clinical diagnoses of heart failure [23]. A gene ontology analysis revealed that the dysregulated genes were mostly related to the development and morphogenesis of cardiac tissue in biological processes (Figure 1H). Notably, the number of downregulated genes was over three-fold larger than that of upregulated genes (Figure 1I), and different cut-offs did not affect the tendency (Appendix A), suggesting that asymmetric changes in gene expression occurred in the heart failure stage. The related results were consistent with previous reports that more transcripts were downregulated rather than upregulated [24,25], which is the hallmark of the mRNA signature in heart failure.

A STRING function enrichment analysis was performed to reveal the interaction of related proteins encoded by dysregulated genes in the heart failure stage after TAC surgery. A total of 40 network nodes were extracted at high interaction score (0.6 or better), indicating functional associations of these proteins with a high confidence (Figure 1J). Interestingly, the nodes encoded by upregulated genes (red) were functionally aggregated in the upper left, including the gold-standard biomarkers (Nppa and Nppb) involved in heart failure diagnosis [23], the molecular markers (Myh7 and Acta1) elevated from the hypertrophic state [26,27], and other cardiac cytoskeleton proteins (Myot and Tpm2) related to heart failure [28,29]. Multiple transcription factors that are critical for cardiac function were enriched in the upper right, including Fos, Jun, Junb, Klf6, and Egr1 [30,31,32], most of which are early response markers activated within 30 min of exposure to hypertrophic stimuli [33]. The lower part contained the nodes related to inflammation and the cell junction, including Tlr4, a key immune receptor that is increased in heart failure and that plays a critical role in myocardial inflammation [34]; Serping1, a serine proteinase inhibitor that reduces inflammation and damage in post-myocardial infraction [35]; and Cdh5, a junction protein exhibiting a cardioprotective role in stress-induced hypoxia or ischemia [36]. Taken together, all these results show that TAC surgery-induced heart failure leads to the altered expression of many genes that are critical for multiple aspects of cardiac function.

### 2.2. Large-Scale Upregulated miRNAs Regulate mRNA at Post-Transcriptional Level

Most mRNAs were downregulated rather than upregulated in the heart failure stage, suggesting that additional mechanisms to suppress steady-state levels of mRNAs might be involved. Considering that miRNAs affect protein translation through the repression or degradation of the mRNA transcripts [8], we analyzed the myocardial miRNA expression profiling in quadruplicate, including the samples for which we performed mRNA profiling (Appendix A). A principal component analysis (PCA) showed that the TAC group was distinguished from the sham group, indicating different miRNA expression patterns between these two groups (Appendix A). QPCR doubly confirmed the expression changes for multiple miRNAs, including those that were significantly upregulated after the TAC surgery, as exampled by miR-1a, Let-7f, miR-486a, and miR-24 (Figure 2A), which are highly abundant in heart tissue and play important roles in cardiac function [10,11,37,38,39]. Importantly, among 234 highly expressed miRNAs (count > 100) (Appendix A), 77 miRNAs were significantly upregulated, but only one miRNA was significantly downregulated by the strict cut-off (*p*-value < 0.05, fold change > 1.70) (Figure 2B and Appendix A), indicating that the miRNAs were upregulated on a large scale in cardiomyocytes at the heart failure stage.

Next, we wondered whether the large-scale upregulation of miRNAs was intrinsically linked to gene expression changes in the TAC heart failure model. By combining two miRNA target prediction databases, Miranda and Targetscan, significantly upregulated miRNA-targeted genes were predicted among those relatively highly expressed genes at the principle of base-pairing interactions in the seed region. The results showed that, except for 642 (7.9%) genes, 7521 (92.1%) genes were targeted by upregulated miRNAs (Figure 2C). Moreover, 6955 (85.2%) genes were targeted by at least two sites with different miRNAs (Figure 2D), suggesting that majority of these genes were regulated by multiple miRNAs. These results should not be biased, since they were consistent with the results from a stricter cut-off (fold change > 2.0) (Appendix A). Importantly, compared to genes without miRNA targeting in their 3′UTRs, the miRNA-targeted genes showed a significant shift towards reduced expression, as indicated by the prospective *p*-values (Figure 2E). The reduced expression trend was more prominent with an increase in the number of miRNA-targeting sites (Figure 2F), which was consistent with the analysis with the stricter cut-off (Appendix A). These related results indicate that the large-scale upregulation of miRNAs degraded the mRNA transcripts of targeted genes, which provides evidence to explain the hallmark of the mRNA signature in heart failure [24,25].

To further explore the molecular mechanisms responsible for the large-scale upregulation of miRNAs, we dissected the levels of precursors as well as the key processing proteins for forming mature miRNAs [8]. We found that most of the upregulated miRNAs showed varying degrees of increase in their primary transcripts (Figure 2G), which was accompanied by some of them simultaneously exhibiting a slight increase at the precursor level. Drosha, a nuclear RNase III-type enzyme and the catalytic subunit of the microprocessor that cleaves pri-miRNA into pre-miRNA [8], remained unchanged at the protein level in heart failure cardiomyocytes after TAC surgery (Figure 2H). A similar thing happened in Dicer (Appendix A), which crops pre-miRNA into mature miRNA. All these related findings suggest that the large-scale upregulation of miRNA might result from the pre-transcriptional control of miRNA biogenesis.

### 2.3. Repression of RBFox2 Protein by Multiple Upregulated miRNAs in Heart Failure

Previously, we reported that RBFox2, the key cardioprotective factor, was greatly reduced at the protein level in the whole heart after TAC surgery [14]. Similar results were also shown in isolated cardiomyocytes, with RBFox2 being ~50% reduced at the protein level (Figure 3A,B). In contrast, its mRNA level remained unchanged (Figure 3C). These results, combined with the previous findings from miRNA profiling in Figure 2, suggest that RBFox2 might be regulated by miRNAs at the post-transcriptional level. To further test this hypothesis, we analyzed possible miRNA-targeting sites in the RBFox2 3′UTR region by overlapping Ago2 CLIP-seq binding profiles with predicted sites from the Targetscan database. A total of 21 strong Ago2 binding peaks were found in the long 3′UTR region (5330nt) of the RBFox2 gene, covering 10 binding sites of upregulated miRNAs (Figure 3D and Appendix A) and indicating that more than one miRNA might target the RBFox2 gene.

To further validate the miRNA-targeting site candidates, we designed the luciferase reporters cloned with wild-type or seed mutant core elements from the RBFox2 3′UTR and tested them in Hela cells by cotransfection with the corresponding miRNAs. Finally, five sites were verified by a luciferase assay at the principle of the base-pairing interaction in the seed region, including those targeted by Let-7f (Figure 3E,F), miR-16 (Figure 3G,H), miR-200b (Figure 3I,J), miR-92a (Appendix A), and miR-24 (Appendix A). These miRNAs were significantly upregulated in cardiomyocytes at the heart failure stage after TAC surgery (Appendix A) and have been reported to be critical for the regulation of cardiac function [37,39,40,41,42], demonstrating that RBFox2 is regulated by multiple cardiac critical miRNAs in heart failure.

Among all the identified miRNA sites targeting RBFox2, the Let-7f target site was the most important one, since it was covered by the highest Ago2 peak in the RBFox2 3′UTR region and was highly conserved in multiple species (Figure 3D,E). The results from miRNA profiling revealed that all Let-7 family members, including Let-7f, are highly abundant in cardiomyocytes (Appendix A) [43] and greatly upregulated at the heart failure stage (Appendix A). Moreover, cotransfection of the luciferase reporter vector with other Let-7 members (Let-7a or Let-7c) also significantly reduced the luciferase assay activity, whereas mutants in the seed region reversed the effect (Appendix A). These results suggest that the Let-7f site might be targeted by the entire Let-7 family due to the high similarity in the nucleotide sequence, especially in the seed region [43].

Based on the luciferase assay results of 16 miRNA binding site candidates in the RBFox2 3′UTR region (Appendix A), a further analysis was performed to examine whether or not species conservation and the Ago2 binding property contributed to the miRNA site determination. All five proven sites were overlapped by Ago2 CLIP-seq peaks, and four of them had highly conserved seed regions in multiple species. (Figure 3K). On the other hand, one of eight poorly conserved candidates was proven to be correct, while none of the candidates (zero out of six) without Ago2 binding turned out to be true. These results suggest that, in addition to species conservation, the Ago2 binding property has a great impact on determining authentic miRNA target sites [44].

### 2.4. Regulation of RBFox2 Protein and Its Downstream Functions by miRNAs

Next, we investigated how these RBFox2-targeted miRNAs affect the RBFox2 protein and its downstream functions in neonatal rat cardiomyocytes (NRCMs). At least a 78-fold increase in the corresponding miRNAs (Let-7f, miR-200b, or miR-16) was detected in cultured NRCMs after three days of transfection (Appendix A), indicating the stable overexpression of the miRNA mimics. In this case, the overexpression of corresponding miRNAs resulted in a ~50% reduction in the RBFox2 protein (Figure 4A and Appendix A), which is comparable to previous results from the heart failure stage in the TAC model (Figure 3A), whereas direct siRNA interference could repress ~80% of the RBFox2 protein. The QPCR results revealed that the RBFox2 mRNA remained unchanged after the targeted miRNA overexpression, while the RBFox2 RNAi directly reduced mRNA production (Appendix A). These results suggest that the repression of the RBFox2 protein by targeted miRNAs was mainly at the post-transcriptional level, which is consistent with previous data (Figure 3A,B).

In a previous report, we discovered that RBFox2 regulates the alternative splicing of numerous genes related to cardiac function and disease [14]. Here, we checked whether or not the overexpression of miRNAs targeting RBFox2 affected downstream alternative splicing functions. According to the RBFox2 CLIP-seq results from mouse cardiomyocytes, RBFox2 bound multiple evolutionarily conserved (U)GCAUG motifs (red) located in the downstream intronic regions of alternative exon 13 of *Enah* (Figure 4B), a validated RBFox2-regulated gene [14]. The overexpression of miRNAs resulted in the partial repression of the alternative exon 13 of *Enah*, while the RBFox2 knockdown by RNAi greatly suppressed the exon. A mild induction of exon 4 alternative splicing upon the miRNA overexpression occurred on *Sorbs2*, another validated RBFox2-regulated gene [14], which is consistent with specific RBFox2-binding events at upstream intronic regions of the exon, as evidenced by CLIP-seq data (Appendix A). Considering that both Enah and Sorbs2 are essential for maintaining the structural integrity of the contractile apparatus in cardiomyocytes [45,46], all related evidence demonstrates that the overexpression of miRNAs targeting *RBFox2* could induce critical cardiac functional genes to undergo RBFox2-dependent alternative splicing changes in NRCMs.

On the other hand, RBFox2 has been found to play a critical role in maintaining normal E–C coupling by protecting *Jph2* from miRNA targeting in the mouse heart [16]. Here, we found that the level of Jph2 protein was reduced by ~50% compared to the negative control after the overexpression of miRNAs targeting RBFox2 (Figure 4A and Appendix A), which is consistent with the results from the RBFox2 knockdown, indicating that the repression of RBFox2 could induce Jph2 protein down-regulation in NRCMs. The RT-PCT results show that the Jph2 mRNA remained unchanged after the RBFox2-targeted miRNA treatment as well as in the RBFox2 knockdown (Appendix A), suggesting that the regulation of Jph2 by RBFox2 is mainly at the post-transcriptional level, which is consistent with our previous report [16]. To further check the E–C coupling function in NRCMs, the calcium transient was examined through the high-resolution line-scan mode in confocal microscopy. The imaging results revealed that calcium transients were attenuated upon the overexpression of RBFox2-targeted miRNAs as well as in the RBFox2 knockdown (Figure 4C and Appendix A), as manifested by greatly reduced amplitudes (Figure 4D,E and Appendix A) and an increased FHDM (Figure 4F and Appendix A). Furthermore, a significant increase in COV during the elevation of calcium transients indicated that desynchronized calcium release occurred in this case (Appendix A). These related results reveal that miRNA-induced RBFox2 protein suppression would cause E–C coupling defects by reducing the Jph2 protein in NRCMs.

### 2.5. Rescuing RBFox2 Protein as Well as Cardiac Function by Antagomir Cocktail

Based on the importance of binding sites as well as the abundance of targeted miRNAs, a cocktail of antagomirs designed according to Let-7 family members was used for miRNA inhibition. Incubation with 100 μM of isoproterenol (ISO) for 6 days resulted in a dramatic induction of heart failure markers (*Nppa*, *Nppb* and *Tlr4*) (Figure 5A) and a significant reduction in the RBFox2 protein (Figure 5B) in cultured NRCMs, indicating cardiomyocyte dysfunction after a prolonged ISO treatment. Importantly, when the antagomir cocktail was introduced simultaneously with the ISO treatment, the elevation of these heart failure markers and the reduction in the RBFox2 protein was compensated to a level comparable to the negative control (Figure 5A,B). These related results indicate that the inhibition of RBFox2-targeted miRNAs could diminish the ISO-induced upregulation of heart failure markers and rescue the downregulation of the RBfox2 protein. In addition, the inhibition of RBFox2-targeted miRNAs could reverse the ISO-induced dysfunction of critical cardiac gene splicing, as shown by more skipping of the alternative exon 13 of Enah after the ISO treatment (Figure 5C), whereas more inclusion of this exon was induced after the simultaneous transfection of the antagomir cocktail.

Slightly increased Jph2 protein levels were detected after the treatment with the antagomir cocktail alone (Figure 5B), which might have been due to the upregulation of the RBFox2 protein as a result of endogenous miRNA repression. Importantly, Jph2 significantly reduced after the ISO treatment, which could be reversed by the simultaneous transfection of the antagomir cocktail, suggesting that the inhibition of miRNAs targeting RBFox2 could rescue the ISO treatment-induced Jph2 protein down-regulation in NRCMs. In addition, the calcium imaging results revealed that the calcium transients were enhanced after the simultaneous transfection of the antagomir cocktail compared with the ISO treatment alone (Figure 5D), which was manifested by a significant increase in the amplitude (Figure 5E) and a slight decrease in the FHDM (Figure 5F). These related results suggest that the inhibition of miRNAs targeting RBFox2 rescued the E–C coupling defects induced by a long-time ISO treatment.

Moreover, the inhibition of RBFox2-targeted miRNAs would also recover dysfunction in NRCMs caused by a long-time (6-day) treatment with phenylephrine (PE) (50 μM). There is evidence that, in addition to rescuing RBFox2 depression (Appendix A), the antagomir cocktail transfection also restored the suppression of the alternative exon in the Enah gene (Appendix A), downregulated the Jph2 protein (Appendix A), and caused defects in calcium handling (Appendix A). These results reveal that the restoration of the RBFox2 protein by antagomirs might serve as a general approach in the treatment of heart failure, given the important role of the RBFox2 protein in cardiac dysfunction.

## 3. Discussion

A lot of studies in the literature have reported a gene expression profile of the heart from animal models as well as human patients [5,6,7], largely based on the whole heart rather than cardiomyocytes. These observations would be biased because nonmuscle cells comprise ~70% of the total cardiac cells [47], of which the largest proportion is cardiac fibroblasts. Here, we decomposed cardiac tissue and isolated cardiomyocytes using the Langendorff perfusion system prior to conducting downstream sequencing and analyses. Based on our sequencing results (Appendix A), we found that markers of cardiac fibroblasts, such as *S100a4*, *Drr2*, *Fap*, *Vegfd*, and *Cdh11* [48], were all expressed at very low levels, indicating that a large proportion of the nonmuscle cells, particularly cardiac fibroblasts, had been removed.

More than one hundred changed genes were detected in duplicates of the mRNA-seq by a stringent cut-off, including marker genes for heart failure (See Figure 1). These results confirm the hallmark of the heart failure mRNA signature [24,25], which means that many more transcripts are downregulated than upregulated. In addition to the development and morphogenesis of cardiac tissue, a gene ontology analysis of dysregulated genes revealed more terms related to heart failure, including a response to a stimulus (*p*-value 1.1 × 10^−9^), the MAPK cascade (*p*-value 1.5 × 10^−7^), and extracellular matrix organization (*p*-value 9.7 × 10^−7^). A network analysis of proteins decoded from changed genes revealed that 40 key nodes were finally identified that are involved in the cytoskeleton, cell connections, transcriptional regulation, inflammatory responses, etc., and almost all nodes had been reported to be associated with heart diseases. Therefore, these nodes have the potential to serve as markers for the diagnosis of heart failure diseases.

The profiling of miRNA was performed with quadruplicates of the controls and experimental groups to reduce the experimental variation (See Figure 2). The results of the principal component analysis (PCA) confirmed the internal consistency in the experimental or control group. A total of 21 miRNAs with different degrees of changes were validated by RT-PCR, and the correlation with sequencing results was very high (R^2^ = 0.86), which illustrated the reliability of deep-sequencing results. With stringent criteria, we identified 77 significantly upregulated miRNAs, illustrating that the wide-scale upregulation of miRNAs in the heart failure stage is consistent with the results from human disease samples [25]. Besides those well-characterized miRNAs (miR-1a, miR-208a, and miR-22) essential for cardiac function, multiple regulated miRNAs have been implicated in heart failure-related cellular signaling pathways. The elevation of the Let-7 and miR-125 members is involved in various cardiac diseases such as arrhythmias, strokes, and myocardial infarctions [43,48]. MiR-26b and -27b are induced in response to a hypoxic event after heart failure [49]. Both miR-23a and miR-24 were increased in a myocardial ischemic/reperfusion injury and induced myocardial apoptosis [50,51]. In addition, miR-23a was reported to target CX43 to enhance mitophagy [52], while miR-24 impaired E–C coupling by targeting Jph2 [39].

A lot of reports have addressed the inhibitory effects of miRNAs on target mRNA translation in various cardiac diseases [11,12,13,39,40,41,42], whereas the effect of miRNAs on mRNA stability has rarely been mentioned. Here, by combining the global profiling of miRNAs and mRNAs, we found that more than 90% of genes were targeted by significantly upregulated miRNAs, most of which had multiple miRNA-targeting sites. An analysis of the mRNA differential expression compared to those miRNA non-targeted genes revealed the degradation of mRNAs in miRNA-targeted genes, which became more prominent with an increase in the number of miRNA binding sites. The data illustrate that widely upregulated miRNAs in heart failure broadly affect gene expression by stimulating mRNA degradation in addition to inhibiting protein translation. The related results also provide the evidence to explain previously discovered asymmetric changes in gene expression, which is the hallmark of the mRNA signature in heart failure [24,25].

MiRNAs are initially transcribed as long-RNA precursors, called primary miRNAs, which require the RNase III enzyme Drosha in the nucleus to be trimmed into precursor miRNAs [8]. The latter are exported to the cytoplasm, where they are subsequently cropped into mature miRNAs by another RNase III enzyme, Dicer. Our further study found that multiple miRNAs were obviously upregulated at the primary level, indicating that they were enhanced at the transcriptional level, which might be the main reason for the upregulation of mature miRNAs in heart failure. Considering that multiple transcription factors showed significant changes in our report (See Figure 1 and Appendix A), some of them have been reported to regulate miRNAs at the transcriptional level, such as LIN28B- and Tnfrsf10b-repressed Let-7 family [53,54], Cebpb-induced miR-16 transcription [55], E2F1-induced miR-92 production [56], and Zebs (Zeb1 and Zeb2) negatively fed back on the transcription of miR-200b [57]. Thus, expression changes to critical transcription factors may regulate miRNAs by transcriptionally enhancing the production of mature miRNAs, which in turn provides feedback to regulate mRNA degradation or translational repression, and this self-loop feedback formed at the overall level would be a major feature of the heart failure transcriptome.

Our previous reports established RBFox2 as a key cardioprotective factor affecting multifaceted functions of the heart [14,15,16]. However, the specific mechanisms leading to its severe impairment during heart failure remain unrevealed. Here, we found that RBfox2 is downregulated mainly at the protein level, and QPCR confirmed that its mRNA remained unchanged (See Figure 3), which is consistent with high-throughput sequencing results. Thus, taken together with the results of large-scale upregulation of miRNAs during heart failure, we suggest that RBFox2 is regulated by certain miRNAs at the post-transcriptional level.

Given the long 3′ UTR region of the RBFox2 gene, the conventional method to predict miRNA binding sites with prediction programs would only yield too large of a number of candidate cites to be identified subsequently. With the aid of the Ago2 CLIP-seq data from the cardiomyocytes in the mouse heart [16], we focused on detecting the miRNA binding sites within the binding peak of Ago2, which greatly reduced the identification of those unlikely sites [44]. The results of a subsequent analysis also confirmed that the sites outside the Ago2 CLIP-seq binding peaks were basically incorrect. Finally, we identified five correct sites at 10 large binding peaks with luciferase reporters cloned from wild-type or seed mutants. All these sites were targeted by significantly upregulated miRNAs, including Let-7, miR-24, miR-16, miR-92a, and miR-200b, which have been reported to affect cardiac functions in many aspects [39,40,41,42,43,51]. The related results indicate that RBFox2 is regulated by multiple miRNAs that are critical for cardiac function during heart failure.

The introduction of exogenous reporters into Hela (or HEK293) cells to test luciferase-based ratiometric activity is a common approach to test whether or not a specific miRNA is targeted to the 3′UTR region of a particular gene. However, a key limitation of the assay is that it cannot provide evidence to prove the changes in the biological function associated with the target gene. Therefore, further verification of these upregulated miRNAs was performed in cultured neonatal rat cardiomyocytes (See Figure 4). The individual overexpression of these miRNAs would reduce the RBFox2 protein by ~50% without affecting its mRNA level, suggesting that these miRNAs could suppress the RBFox2 protein in cardiomyocytes, which is consistent with the results of the luciferase assay. Furthermore, RBFox2 suppression by these miRNAs could cause splicing changes in Enah and Sorbs2, the key cytoskeleton proteins that maintain normal cardiac contraction [45,46]. The trend of splicing repression is consistent with interference to RBFox2 by siRNA, although the degree of change is relatively mild. Related results confirm that the suppression of the RBFox2 protein by miRNAs affects the alternative splicing of critical cardiac proteins, which is consistent with the previous reports on RBFox2-KO hearts [14].

RBFox2 is a multifunctional protein that plays a cardioprotective role in cardiac tissue. In addition to broadly affecting the alternative splicing of cardiac critical genes, it transcriptionally represses miRNAs as well as competes for miRNA targeting to maintain normal E–C coupling in the heart [15,16]. The most critical downstream gene is Jph2, which is expressed from the embryonic stage in cardiac tissue and maintains normal E–C coupling by interacting with DHPR and RyR2 and preserving the morphology of T-tubules in the mature heart [58]. A deficiency in Jph2 significantly impairs cardiac E–C coupling, which is manifested by a poor synchrony and a reduced amplitude of calcium transient [59], and this effect already emerged in the embryonic stage of the heart after the knockout of Jph2 [60]. Both repression by miRNA and the siRNA interference of RBFox2 could reduce the Jph2 protein significantly without affecting its mRNA level, indicating that RBFox2 can regulate the Jph2 protein expression at the post-transcriptional level. Furthermore, the repression of RBFox2 altered the calcium transient characteristics; in particular, its amplitude was impaired and the FDHM was prolonged, which was consistent with the results of the knockout of RBFox2 in adult mouse cardiomyocytes [14], as well as the siRNA interference of RBFox2 in neonatal rat cardiomyocytes. These related results indicate that the repression of RBFox2 by miRNA causes the downregulation of the Jph2 protein, which in turn impairs the E–C coupling function in both mouse and rat hearts.

Indeed, in addition to muscle cells, RBFox2 is ubiquitously expressed with high abundance in most cell types, including MEF, fibroblasts, embryonic stem cells, and cancer cells [15,61,62]. On the other hand, the RBFox2-targeted miRNAs have been reported to play critical roles in other cells and tissues. The Let-7 family has been found to be highly expressed in many systems and is related to angiogenesis [37]. MiR-24 is associated with tumor initiation and progression [63], while miR-200b plays crucial roles in the suppression of the epithelial–mesenchymal transition (EMT) [57]. MiR-16 was first discovered in chronic lymphocytic leukemia and regulates immune responses, and also functions as a tumor suppressor [64]. MiR-92a is abundant in human MSCs and suppresses angiogenesis by the down-regulation of HGF secretions [65]. Therefore, RBFox2 repression by these miRNAs individually or synergistically might be a universal mechanism that plays important roles in other cells and tissues and is critical for disease progression.

ISO is a synthetic non-selective β-adrenoceptor agonist widely used for inducing heart failure models in rodents [66]. Incubation with ISO for 2–3 days at the cell level can induce cardiomyocytes to develop a hypertrophic phenotype, and a daily injection of ISO for 2 weeks at the animal level is able to cause heart failure [67]. Here, we found that 6 days of high-dose ISO incubation was able to cause the overexpression of the heart failure markers *Nppa*, *Nppb*, and *Trl4*; the significant downregulation of the RBFox2 protein; repression of the alternative splicing in the Enah gene; and the suppression of the Jph2 protein in NRCMs (See Figure 5). However, hypertrophy marker genes such as *Myh7* remained unchanged in the ISO group compared to the control (Appendix A). These results suggest that NRCMs developed to the severe dysfunctional stage mimicking heart failure rather than an early hypertrophic phenotype after a high-dose and prolonged ISO incubation. In addition, the calcium imaging results revealed the significant defects of calcium handling in dysfunctional NRCMs under this condition. The possible mechanism was that the prolonged incubation of ISO induced NRCMs to develop a severe dysfunction resembling a heart failure phenotype, which depressed RBFox2 and led to the repression of its downstream protein Jph2, and then eventually impaired cardiac E–C coupling.

Considering the expression levels of miRNAs targeting RBFox2, the strength of Ago2 binding in its 3′UTR, and the species conservation of the seeds, we attempted to restore the RBFox2 protein by employing miRNA inhibitors (antagomir cocktail) designed according to the Let-7 family members. The introduction of the antagomir cocktail could not only reverse the ISO-induced downregulation of heart failure markers, but also rescue downstream function defects caused by the repression of the RBFox2 protein, as indicated by the restoration of splicing changes in the Enah gene, the recovery of the Jph2 protein, and the enhancement of calcium transients. Similarly, the inhibition of miRNAs targeting RBFox2 also restored the cardiomyocyte dysfunction induced by phenylephrine (PE), a classical α1-adrenoceptor agonist [68]. These results reveal that miRNA inhibitors could rescue the downregulation of cardiac function associated with RBFox2 in dysfunctional cardiomyocytes.

In summary, we found that large-scale upregulation of miRNAs is another hallmark of the heart failure transcriptome and is responsible for the degradation of many genes at the mRNA level. Multiple upregulated miRNAs were confirmed to target RBFox2 in heart failure, which not only suppressed the RBFox2 protein at the post-transcriptional level, but also impaired its protective effects on E–C coupling as well as alternative splicing in cardiomyocytes. The inhibition of Let-7 members, the most critical miRNAs targeting RBFox2, was able to rescue the RBFox2 protein and restore its downstream effects in dysfunctional cardiomyocytes. All these results not only emphasize the protective role of RBFox2 in cardiac function, but also demonstrate its therapeutic potential for heart failure. The inhibition of miRNAs targeting the RBFox2 protein may serve as an important mechanism for the development of useful drugs for the treatment of heart failure. Further studies will be required at the animal level, focused on restoring the RBFox2 protein and exploring disease treatments related to heart failure.

## 4. Materials and Methods

### 4.1. Transverse Aortic Constriction and Cardiomyocytes Isolation

Male mice (2-month-old) were anesthetized with ketamine (100 mg/kg, ip)/xylazine (5 mg/kg, ip) through a pressure overload to perform the TAC operation as previously described [69]. Echocardiography examinations were measured before surgery and 14 weeks post-TAC. Heart tissue from the sham or TAC-treated 14-week-old mice was disaggregated by a Langendorff perfusion system to isolate the cardiomyocytes prior to downstream experiments. Those isolated cardiomyocytes from one mouse heart were equally divided into 4 separate tubes for the following Western blot, RNA-seq, miRNA-seq, or RT-PCR procedures.

### 4.2. Isolation and Culture of NRCMs

Two-day-old Sprague–Dawley rats were anesthetized with pentobarbital sodium, and the hearts were quickly removed and digested with an enzyme solution containing collagenase type B, collagenase type D, and protease type XIV. When the heart became swollen and hard, it was cut into several chunks and further digested in a shaker (60–70 rpm) for 10 min at 37 °C in the same enzyme solution. The supernatants were filtered and gently centrifuged, and then the pellets were resuspended in Ca2^+^ solution I (125 mM). After depositing for 10 min, the supernatants were aspirated and the myocytes were resuspended in Ca2^+^ solution II (250 mM). The final cell pellet was suspended in Ca2+ solution III (500 mM). Meanwhile, the shake–harvest procedure was repeated several times until all the chunks were digested.

After isolation, the NRCMs were cultured in DMEM supplemented with 10% heat-inactivated FBS and 1% penicillin/streptomycin at 37 °C in a humidified atmosphere of 5% CO^2^ and 95% air.

### 4.3. Luciferase Assay

For the luciferase assay, reporters were constructed by cloning the PCR-amplified core elements of mouse RBFox2 3′UTR into the pmirGLO vector between Xho I and Sal I (IGE). Hela cells were transfected with 100 nmol of miRNA mimics (GenePharma, Suzhou, China) and 200 ng of reporters with Lipofectamine 2000 (Thermofisher, Waltham, MA, USA, 11668019) for 48 h. To examine the effect of miRNAs on RBFox2, the cells were lysed with 1× passive lysis buffer, and luciferase assays were performed according to the Dual Luciferase Assay System protocol (Promega, Madison, WI, USA, E1910) using SpectraMax Id3 microplate readers (Molecular Devices, San Jose, CA, USA).

### 4.4. Transfection of miRNA and Antagomirs

For siRNA interference and miRNA overexpression, NRCMs were transfected with siRNA or the miRNA mimic using R-Fect (Baidai, Changzhou, China) according to the manufacturer’s instructions. Western blotting and functional studies were carried out 3 days after transfection. siRNA was synthesized by Gene Pharma (Suzhou, China), and the sequences of siRNA and mimic are listed (Appendix A).

For preparing the antagomir cocktail, antagomirs designed against Lef-7a/b/c/d/e/f/g-5p (GenePharma, Suzhou, China) were mixed at the same concentration. NRCMs were transfected with 200nM of the antagomir cocktail using R-Fect (Baidai, Changzhou, China) according to the manufacturer’s instructions and harvested for further functional studies after incubating for 6 days.

### 4.5. Western Blot

Isolated mouse cardiomyocytes or NRCMs were lysed with a lysis buffer supplemented with a protease inhibitor cocktail and PMSF to prepare the total protein extracts. The protein samples were separated by SDS-PAGE and transferred into nitrocellulose membranes, which were blocked in 5% non-fat milk for 1 h at room temperature followed by incubating with primary antibodies overnight at 4 °C. Then, the samples were incubated with HRP-conjugated anti-mouse or anti-rabbit secondary antibodies (7076S; 7074S; Cell Signaling Technology, Denvers, MA, USA) for 1 h at room temperature. Immunoblot signals were incubated with ECL Advance Western blotting detection reagents (Thermo Scientific, Waltham, MA, USA, 34577) and detected by Tanon (6100C). All the protein levels were normalized by GAPDH. The following primary antibodies were used in the present study: anti-RBFox2 (Bethyl, Montgomery, TX, USA, A300-864A); anti-Jph2 (SantaCruz, Dallas, TX, USA, sc-377086); anti-Dorsha (SantaCruz, Dallas, TX, USA sc-393591); anti-Dicer (SantaCruz, Dallas, TX, USA sc-136979); and anti-GAPDH (FuDe, Wuhan, China, BM3874).

### 4.6. RNA-seq and miRNA-seq

In each sample, >1 × 10^5^ of cardiomyocytes (~1/4 of the total myocytes from one mouse heart) were used to extract RNA for deep sequencing experiments. An amount of 2–4 ug of total RNA was extracted from each sample using the TRIzol^®^ reagent (Invitrogen, Waltham, MA, USA). The RNA integrity number (RIN) was measured by GENE DENOVO and the samples with RIN > 8.0 were processed for deep sequencing. Finally, a total of 8 samples (1 ug each) from 4 pairs of sham and TAC mice were used to perform the miRNA-seq (GENE DENOVO, Guangzhou, China), which included the 4 samples (500 ng each) for RNA-seq (GENE DENOVO, Guangzhou, China).

### 4.7. QPCR for mRNA and miRNA

The total RNA from cardiomyocytes or NRCMs was extracted using the TRIzol^®^ reagent (Thermo Scientific, Waltham, MA, USA, 15596026). Reverse transcription reactions were performed using a Superscript^®^-III kit ((Thermo Scientific, Waltham, MA, USA, A10752030) with a total RNA amount of 5 μg according to the manufacturer’s instructions. QPCR was performed with TB Green Premix Ex TaqII (Takara, Kusatsu, Japan, RR820) using a real-time PCR machine (Analytikjena, Jena, Germany), and *GAPDH* was used as an endogenous control. The primers were designed by primer 6, and are listed in Appendix A.

The procedure for primary miRNA detection was the same as for mRNA. For mature miRNAs and precursor miRNAs, reverse transcription reactions were performed using a Mir-X miRNA First-Strand Synthesis kit (Takara, Kusatsu, Japan, 638313). QPCR was performed with TB Green Advantage qPCR Premix (Takara, Kusatsu, Japan, 639676). Two snoRNAs (Sno135 and Sno142) were used as endogenous controls. Mature miRNA primers were designed from the DNA version of mature miRNA; precursor miRNA primers were designed from individual precursor miRNAs (outside the mature miRNA region). All the primers are listed in Appendix A.

### 4.8. Calcium Imaging

NRCMs were seeded in a 12-well plate with a glass bottom and loaded with Cal-590 (AATbio, Pleasanton, CA, USA, 20510) for 7 min at 37 °C, followed by rinsing 3 times with DMEM medium. Then, the cells were incubated in phenol red-free DMEM medium in a 37 °C heated chamber. Calcium transients were imaged in line-scan mode using a Nikon confocal microscope with a 40× oil objective. To eliminate transient effects on calcium signaling by ISO (or PE), the solution was replaced with normal DMEM medium 2 h before calcium imaging for those ISO or PE treatment groups.

### 4.9. Data Analysis

Statistical analyses were performed using Student’s *t*-test. A bioinformatics analysis was performed using the OmicShare tools on www.omicshare.com/tools, accessed from 14 May 2022 to 24 May 2022. The prediction of miRNA targets was performed by TargetScan (http://www.targetscan.org/, accessed on 23 May 2022). Multiple alignments of 30 species and measurements of evolutionary conservation results were exported from a genome browser database (http://genome.ucsc.edu/, accessed on 24 June 2022). The Ago2 and RBFox2 CLIP-seq data was from our previous reports ([14,16]). The species conservation, Ago2, and RBFox2 binding intensity were generated by using igvtools (http://www.broadinstitute.org/software/igv/home, accessed on 24 June 2022).

The mRNA-seq from the sham and TAC group were normalized by total counts. A total of 8163 relatively highly expressed genes (FPKM > 2) were used for the Volcano plot and heatmap, and acted as background genes in the GO enrichment analysis. A functional protein association network was analyzed by a string program (http://www.string-db.org, accessed on 16 May 2022) based on changed genes by a stricter cut-off (FPKM > 5, fold change > 2, and *p*-value < 0.05). The network nodes represent proteins and the edges represent the predicted functional associations; red and green represent increased and decreased expression, respectively; the circle size indicates the number of protein interactions; and the confidence score is the approximate probability that a predicted link exists between two enzymes in the same metabolic map in the KEGG database. A high confidence (>0.6) was used in this analysis.

The results of the miRNA-seq were normalized by two unaltered miRNAs (miR-126a-3p and miR-19b-3p), which were verified by miRNA QPCR. Genes with miRNA binding and unbinding are plotted in a cumulative fashion, the cumulative probability was calculated, and the figure was performed by sigma plot; the statistical significance was determined by a Kolmogorov–Smirnov (KS) test.

## Figures and Tables

**Figure 1 ijms-24-01283-f001:**
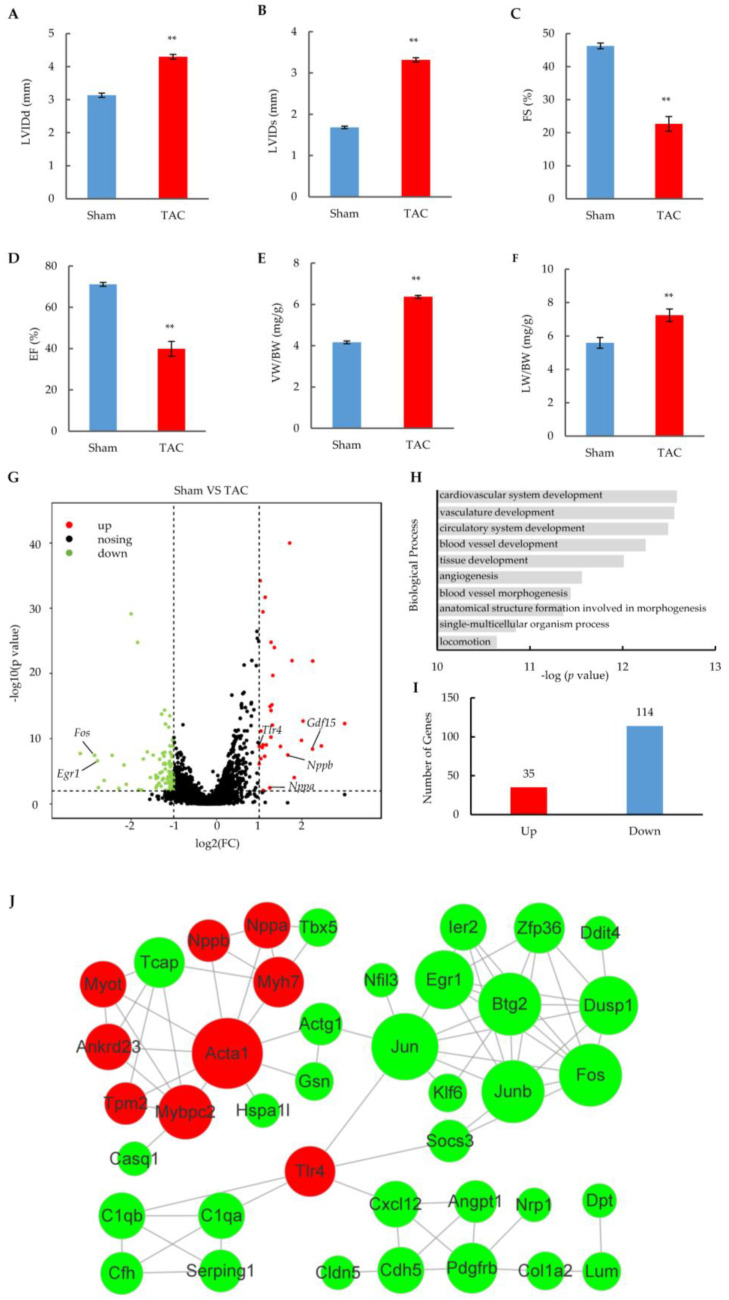
Heart failure phenotypes and cardiomyocyte transcriptome alterations induced by TAC surgery. (**A**–**D**) Key echocardiography parameters of sham and TAC mice after 14 weeks of surgery. LVIDd: left ventricular internal dimension values at end-diastole; LVIDs: left ventricular internal dimension values at end-systole. The results are from 6 pairs of sham and TAC mice and are shown as means ± SEM; ** *p* < 0.01. (**E**,**F**) Ventricle weight (LW) and lung weight (LW) ratio to body weight (BW) in sham or TAC mice after 14 weeks of surgery. The results are shown as means ± SEM; *n* = 6; ** *p* < 0.01. (**G**) Volcano plot of gene expression in cardiomyocytes of TAC mice compared with sham-operated mice. Results are means of duplicate datasets. The red and green dots represent significantly upregulated and downregulated genes, respectively, with the cut-off at fold changes > 2 and a *p*-value < 0.05. Unchanged genes are represented in black. Annotated genes represent heart failure biomarkers. (**H**) Gene ontology analysis of dysregulated genes enriched in biological processes. Data represent the top ten GO terms with the lowest *p*-value. (**I**) Number of significantly induced and repressed genes in cardiomyocytes after TAC surgery. (**J**) Functional protein association networks of related proteins encoded by dysregulated genes with high confidence (>0.6). Red and green nodes represent significantly upregulated and downregulated proteins, respectively. The size of nodes indicates the strength of association.

**Figure 2 ijms-24-01283-f002:**
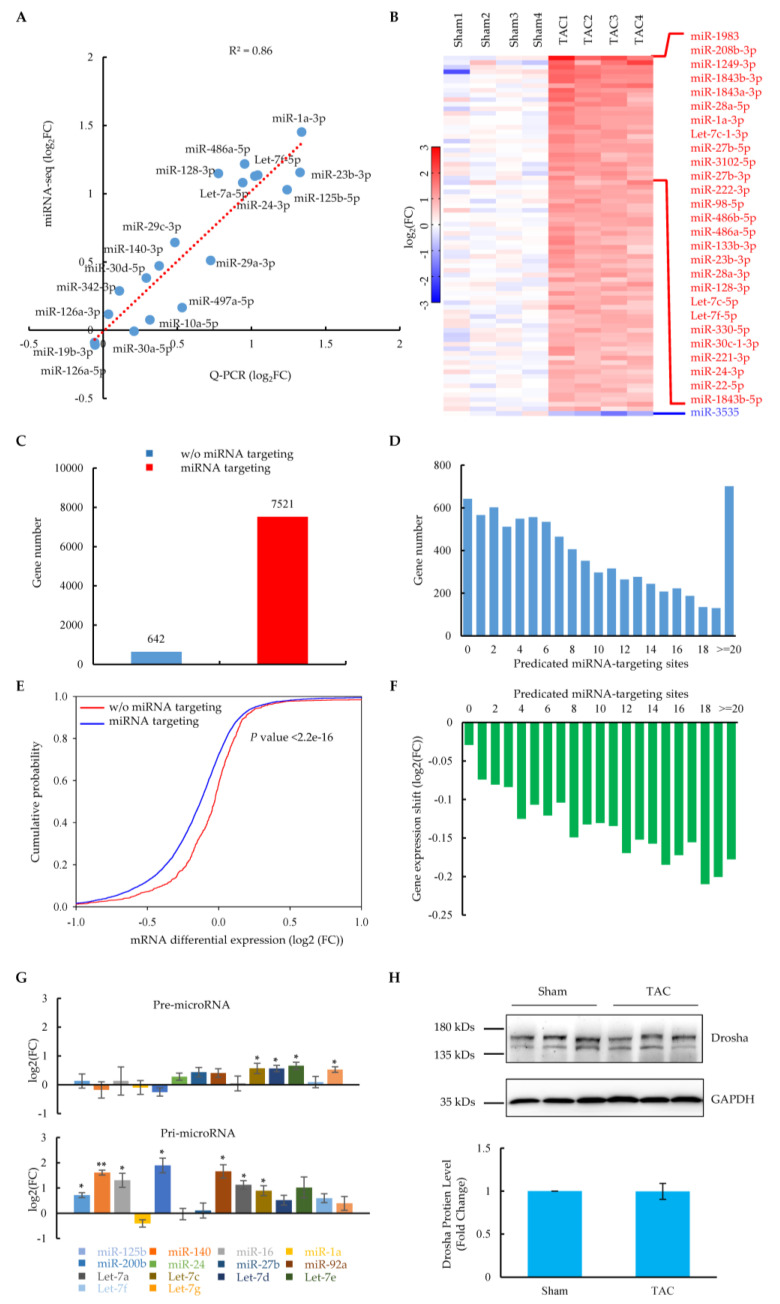
Signature of miRNA expression and their regulation on mRNAs in cardiomyocytes at the heart failure stage. (**A**) QPCR validation of the expression of 19 miRNAs from miRNA global profiling. The red dotted line shows the correlation of QPCR results with miRNA-seq. (**B**) Heatmap of significantly changed miRNAs (*p*-value < 0.05, fold change > 1.70) from miRNA-seq (in quadruplicate) in cardiomyocytes of sham and TAC mice. Results were sorted by the average of the fold change. Red indicates increased expression; blue indicates decreased expression. The 27 most increased and one decreased miRNAs are highlighted on the right. (**C**) Predicted genes targeted by significantly altered miRNAs in cardiomyocytes. (**D**) Distribution of miRNA target genes versus predicated targeting sites. (**E**) Functional correlation between miRNA targeting and gene expression. Differential expression of mRNAs with or without predicated miRNA targeting was plotted in a cumulative fashion. Statistical significance was determined by KS test. (**F**) Distribution of gene expression shift versus predicated miRNA-targeting sites. (**G**) QPCR measurement of relative levels of precursor (upper panel) and primary (bottom panel) miRNAs in sham and TAC cardiomyocytes (*n* = 3 pairs of sham and TAC mice). Values are shown as means ± SEM. * *p* < 0.05; ** *p* < 0.01. (**H**) Western blotting analysis of Drosha protein expression in sham and TAC cardiomyocytes (*n* = 3 pairs of sham and TAC mice); values are shown as means ± SEM.

**Figure 3 ijms-24-01283-f003:**
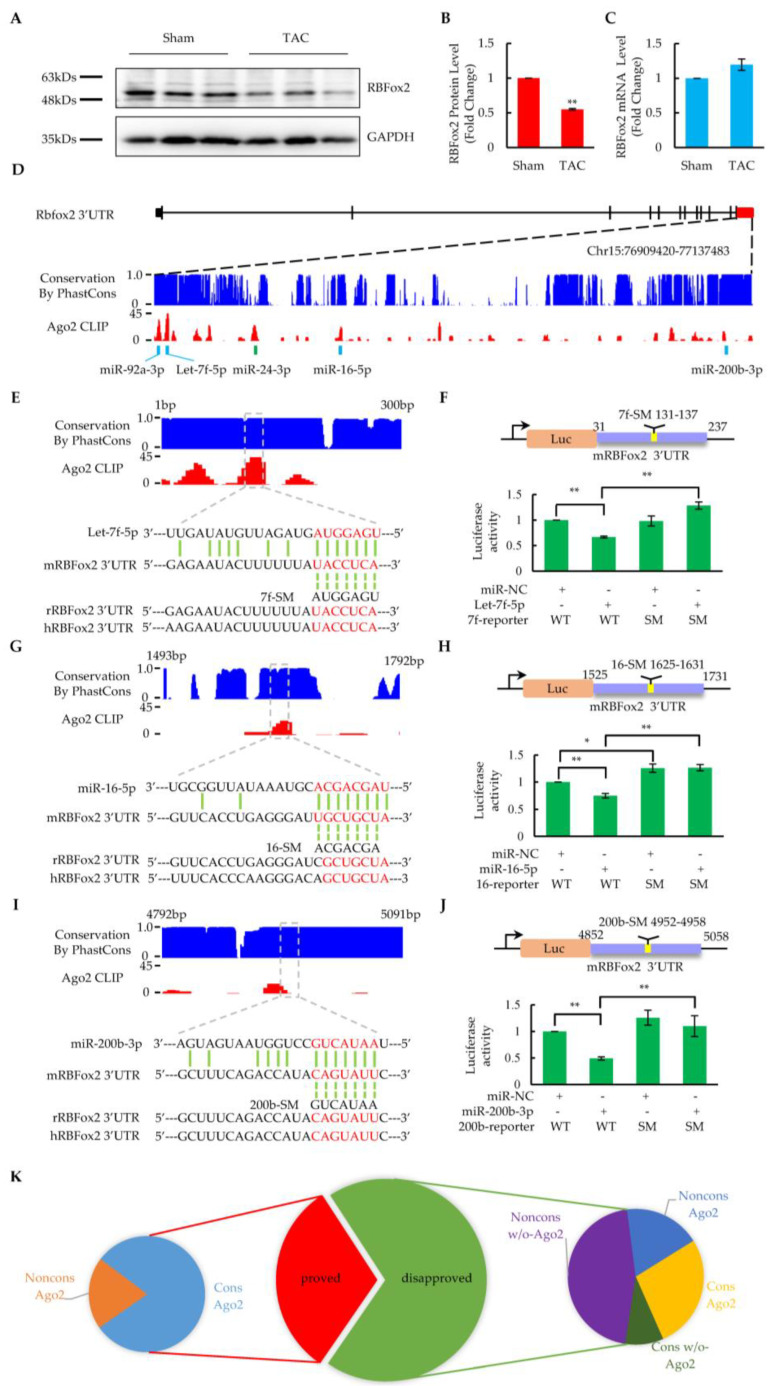
Identification of critical miRNAs targeting to RBFox2. (**A**) Western blotting analysis of RBFox2 protein in the sham and TAC cardiomyocytes (*n* = 3 pairs of sham and TAC mice). (**B**) Relative fold change of RBFox2 protein normalized to GAPDH. Values are shown as means ± SEM; ** *p* < 0.01. (**C**) QPCR analysis of RBFox2 mRNA expression in the sham and TAC cardiomyocytes (*n* = 6). Values are shown as means ± SEM. (**D**) Ago2 binding events in the 3′UTR of mouse RBFox2. The blue bar charts represent species conservation and the red bar charts represent Ago2 binding intensity. Five validated sites with corresponding miRNAs are highlighted below the panel. (**E**) Let-7f-5p targeting site in the 3′UTR of mouse RBFox2. Species conservation (blue) and Ago2 binding intensity (red) are represented in the upper panel. Predicted targeting regions are highlighted in the dashed boxes. The lower panel shows the base pairing between the miRNA and its target site, the mutations introduced in the seed regions, and the corresponding sequences in rats and humans. (**F**) Results of luciferase assays in Let-7f-5p targeting site, corresponding to (**E**). miR-NC: miRNA negative control; WT: luciferase reporter with wild-type core elements from mouse RBFox2 3′UTR; SM: luciferase reporter with seed mutant core elements. Values are shown as means ± SEM. *n* = 5 for each group; ** *p* < 0.01. (**G**) MiR-16-5p targeting site in the 3′UTR of mouse RBFox2. The bar charts and base pairing information are similarly annotated as in (**E**). (**H**) Results of luciferase assays in miR-16-5p targeting site, corresponding to (**G**). Values are shown as means ± SEM. *n* = 4 for each group; * *p* < 0.05; ** *p* < 0.01. (**I**) MiR-200b-3p targeting site in the 3′UTR of mouse RBFox2. The bar charts and base pairing information are similarly annotated as in (**E**). (**J**) Results of luciferase assays in miR-200b-3p targeting site, corresponding to (**I**). Values are shown as means ± SEM. *n* = 4 for each group; ** *p* < 0.01. (**K**) Species conservation and Ago2 binding property of predicted miRNA-targeting sites in the 3′UTR of mouse RBFox2. Cons: high conservation across species in seed region (score > 0.6); Noncons: Non-conservation across species (score < 0.6), Ago2: a site overlapping with the Ago2 peak (s); w/o-Ago2: a site that is not overlapping with the Ago2 peak.

**Figure 4 ijms-24-01283-f004:**
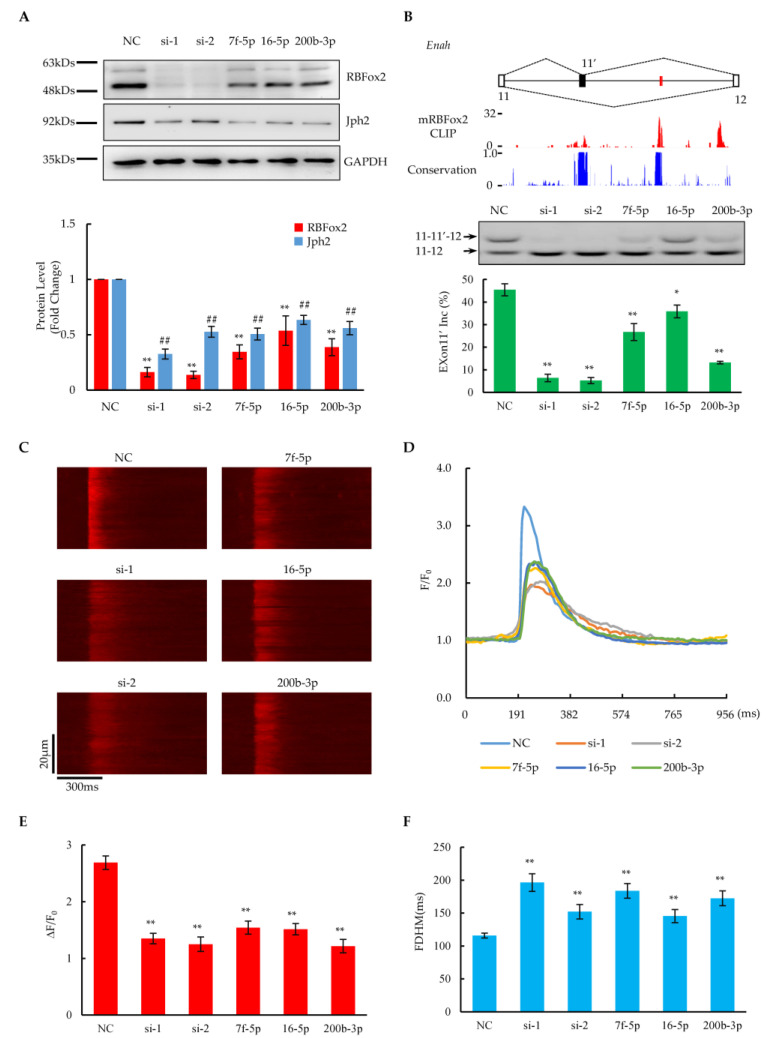
Effect of miRNAs on RBFox2 protein and its related biological functions. (**A**) Upper panel shows Western blotting analysis of RBFox2 and Jph2 protein levels after transfection with RBFox2-targeted miRNA or RBFox2 siRNA in NRCMs, respectively. Bottom panel shows relative fold changes in RBFox2 and Jph2 protein levels normalized to GAPDH. Values are shown as means ± SEM. NC: negative control; Si: RBFox2 siRNA; 7f-5p: Let-7f-5p; 16-5p: miR-16-5p; 200b-3p: miR-200b-3p. *n* = 3 for each group; ** *p* < 0.01 for RBFox2; ## *p* < 0.01 for Jph2. (**B**) Splicing changes in alternative exon 11 of Enah. Top: The gene structure illustrating the alternative exon. The red lines indicate the locations of conserved (U)GCAUG motifs. Middle: The bar charts of RBFox2 CLIP-seq signals from mouse cardiomyocytes and conservation scores across species. Bottom: The PCR results and analysis percentages of exon 11′ inclusion in each sample. Values are shown as means ± SEM. *n* = 4 for each group; * *p* < 0.05, ** *p* < 0.01. (**C**) Confocal images of spontaneous calcium transients in NRCMs after transfection with miRNAs or siRNAs. (**D**) Spatially averaged traces of calcium transient, corresponding to (**C**). (**E**,**F**) Statistics of amplitude (ΔF/F0) and full duration at half-maximum (FDHM) of calcium transients. The data are expressed as means ± SEM. *n* = 43 for NC, *n* = 21 for si-1, *n* = 20 for si-2, *n* = 17 for 7f-5p, *n* = 28 for 16-5p, and *n* = 22 for 200b-3p; ** *p* < 0.01.

**Figure 5 ijms-24-01283-f005:**
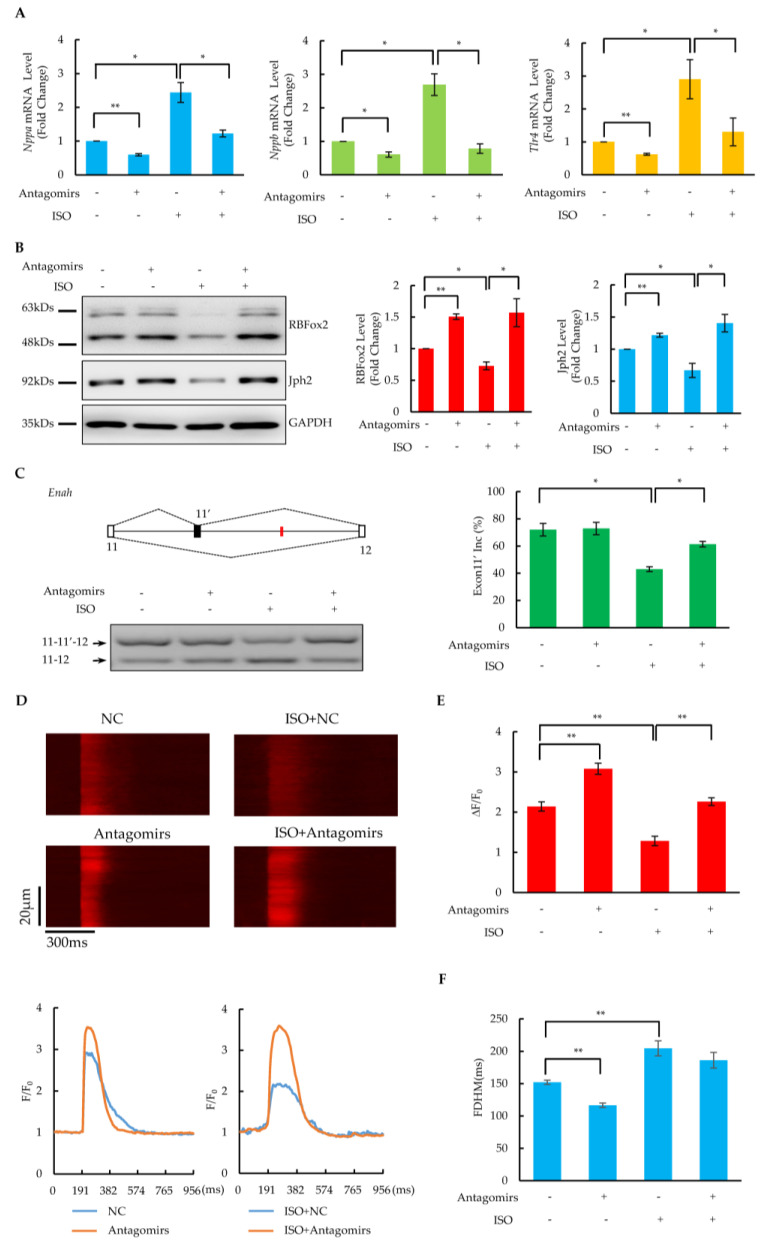
Rescue of RBFox2 protein and its downstream functions in dysfunctional NRCMs by antagomirs. (**A**) The effect of the antagomir cocktail on the mRNA expression of *Nppa*, *Nppb*, and *Tlr4* in NRCMs after ISO treatment. Values are shown as means ± SEM. *n* = 3 for each group; * *p* < 0.05, ** *p* < 0.01. (**B**) The effect of the antagomir cocktail on the RBFox2 and Jph2 proteins in NRCMs after ISO treatment. Left panel: Western blotting analysis of RBFox2 and Jph2 proteins. Right panels: relative fold changes in RBFox2 and Jph2 protein levels normalized to GAPDH. Values are shown as means ± SEM. *n* = 3; * *p* < 0.05, ** *p* < 0.01. (**C**) Splicing changes in alternative exon 11′ of Enah. Left panel: The gene structure illustrates the alternative exon and the red lines indicate the locations of conserved (U)GCAUG motifs. PCR results are shown at the bottom. Right panel: analysis percentages of exon 11′ inclusion in each sample. Values are shown as means ± SEM. *n* = 3; * *p* < 0.05. (**D**) Confocal images of spontaneous calcium transients in NRCMs after ISO treatment or (and) antagomir cocktail transfection. Spatially averaged traces of calcium transient in different treatments are shown in the lower panel. (**E**,**F**) Statistics of amplitude (ΔF/F0) and full duration at half-maximum (FDHM) of calcium transients. The data are expressed as means ± SEM. *n* = 13 for NC, *n* = 20 for antagomirs, *n* = 16 for ISO, *n* = 18 for ISO + antagomirs; ** *p* < 0.01.

## Data Availability

The data presented in this study are available in the article. The mRNA-seq and miRNA-seq data presented in this report are available at the Gene Expression Omnibus under the accession number GSE215303.

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
