# Peer review of "Suppression of RBFox2 by Multiple MiRNAs in Pressure Overload-Induced Heart Failure"

_ijms, 2023, doi:10.3390/ijms24021283_

Round 1

Reviewer 1 Report

In this study, Gu et.al. investigated the role of RBFox2 and its molecular mechanism in pressure-overload induced heart failure. Authors first performed transcriptomic analysis including mRNAs and miRNAs in isolated cardiomyocytes and identified differentially expressed genes and related miRNAs. Second, followed up by their previous studies, authors identified let-7 members most critical miRNAs targeting to RBFox2, was able to rescue RBFox2 protein and restore its downstream effects in ISO and PE treated NRCMs.

Overall, this study has a good rationale and clear hypothesis, detailed method section and appropriate references. However, it can be further improved by addressing the comments below:

Major comments: 

1.     How many isolated cardiomyocytes were used to perform RNA seq? Is there a difference in how many cells isolated from Sham and TAC group? This information is missing in the method section.

2.     Figure 1H only showed the five GO terms with lowest p-value, how about GO terms related with oxidative phosphorylation, mitochondrial dysfunction or extracellular matrix formation which are known to be related with heart failure, with just the 5 GO terms in 1H, it is hard to predict the accuracy of your analysis 

3.     Your 2015 cells report paper (PMCID: PMC4559494) showed that RBFox2 protein was largely diminished in 5-weeks post TAC heart, expression level of RBFox2 is associated with heart failure, in figure 3A, the reduction in RBFox2 protein expression is around 50% 14 weeks post TAC, how to explain this discrepancy? 

4.     How many mice were used to perform the miRNA analysis? Text said included 2 samples from RNA seq, what is the reason of using different number of animals for these two experiments?

5.      Figure 5 used 100 μM Isoproterenol (ISO) treatment for 6 days, the concentration is quite high and treatment time is very long, is there a lot of cardiomyocyte apoptosis? At lower ISO concentration and 48 hours treatment can already induce folds of increase in ANP and BNP, what is the reason of choosing this does and time point? 

6.     Isoproterenol is known to enhance the influx of Ca2+, in Figure 5D and E, ISO treatment induced the opposite effect, is this correct or it is a compensatory effect?

7.     Dose and time of phenylephrine (PE) treatment is missing. 

8.     Title is overstating the findings from current study

Minor comments:

1.     Figure 1G, Y axis matrix can be adjusted to show more clear distribution of DEGs.

2.     Line 113-114, text wrote 149 genes showed significant change in TAC group by a strict cut-off (P <0.05; fold change>2.0), however, supplemental figure 1D showed 149 genes are detected with a cut-off of fold change is >1.5 which is not consistent. 

3.     Supplemental figure 1D and 1E, which one is fold change >1.5 and which is fold change >2.0

4.     Line 428, fibrosis is a pathological remodeling event in the heart, not a cardiac disease. 

5.     Line 311-315, repeated sentences. 

6.     Line 313, 395, 544,564, typo

Author Response

Thank you very much for your hard work in reviewing our manuscript. We have carefully read the comments and provided our point-by-point responses. Please see the attachment file for details.

Reviewer 2 Report

RNA binding protein fox-1 homolog 2 (RBFox2) is a regulatory protein of
splicing events by binding to 5'-UGCAUGU-3' elements and by preventing
binding of U2AF2 to the 3'-splice site. It regulates alternative
splicing of tissue-specific exons, holding critical developmental functions
in many tissues including the heart. Its mutations has been associated with
hypoplastic left heart syndrome [ref#19]. Also, it has been associated with
heart failure (ref#16). The authors further explored this particular notion
in experimental generated heart failure model in mice after TAC surgery.
They obtained the transcriptomic changes and in particular miRNA changes
in mouse failing hearts compared to controls. They elegantly showed
the association of particular miRNA changes with RBFox2 by repressing it.
Overall it is a well constructed experimental study, suggesting interesting
RBFox2 and miRNA targets for the treatment of heart failure.

Author Response

Thank you very much for your appreciation and efforts in reviewing our manuscript.